# A Rejection Gene Expression Score in Indication and Surveillance Biopsies Is Associated with Graft Outcome

**DOI:** 10.3390/ijms21218237

**Published:** 2020-11-03

**Authors:** Betty Chamoun, Anna Caraben, Irina B. Torres, Joana Sellares, Raquel Jiménez, Néstor Toapanta, Ignacio Cidraque, Alejandra Gabaldon, Manel Perelló, Ricardo Gonzalo, Francisco O’Valle, Francesc Moreso, Daniel Serón

**Affiliations:** 1Nephrology Departments, Hospital Universitari Vall d’Hebron, 08035 Barcelona, Spain; md.chamounbetty@gmail.com (B.C.); Annacaraben@gmail.com (A.C.); ibtorres@vhebron.net (I.B.T.); jsellares@vhebron.net (J.S.); racheljiga@hotmail.com (R.J.); ntoapanta@vhebron.net (N.T.); ignacio.cidraque@vhir.org (I.C.); mperello@vhebron.net (M.P.); dseron@vhebron.net (D.S.); 2Pathology Departments, Hospital Universitari Vall d’Hebron, 08035 Barcelona, Spain; agabaldon@vhebron.net; 3Statistics and Bioinformatics Unit (UEB), Vall d’Hebron Research Institute (VHIR), 08035 Barcelona, Spain; ricardo.gonzalo@vhir.org; 4Pathology Department, Instituto de Biopatología y Medicina Regenerativa (IBIMER), Instituto de Investigación Biosanitaria (ibs.GRANADA), University of Granada, 18010 Granada, Spain; fovalle@ugr.es; 5Department of Medicine, Autonomous University of Barcelona, 08035 Barcelona, Spain

**Keywords:** renal transplantation, biopsies, rejection, transcriptomics, microarrays, borderline changes, interstitial fibrosis and tubular atrophy

## Abstract

Rejection-associated gene expression has been characterized in renal allograft biopsies for cause. The aim is to evaluate rejection gene expression in subclinical rejection and in biopsies with borderline changes or interstitial fibrosis and tubular atrophy (IFTA). We included 96 biopsies. Most differentially expressed genes between normal surveillance biopsies (*n* = 17) and clinical rejection (*n* = 12) were obtained. A rejection-associated gene (RAG) score was defined as its geometric mean. The following groups were considered: (a) subclinical rejection (REJ-S, *n* = 6); (b) borderline changes in biopsies for cause (BL-C, *n* = 13); (c) borderline changes in surveillance biopsies (BL-S, *n* = 12); (d) IFTA in biopsies for cause (IFTA-C, *n* = 20); and (e) IFTA in surveillance biopsies (IFTA-S, *n* = 16). The outcome variable was death-censored graft loss or glomerular filtration rate decline ≥ 30 % at 2 years. A RAG score containing 109 genes derived from normal and clinical rejection (area under the curve, AUC = 1) was employed to classify the study groups. A positive RAG score was observed in 83% REJ-S, 38% BL-C, 17% BL-S, 25% IFTA-C, and 5% IFTA-S. A positive RAG score was an independent predictor of graft outcome from histological diagnosis (hazard ratio: 3.5 and 95% confidence interval: 1.1–10.9; *p* = 0.031). A positive RAG score predicts graft outcome in surveillance and for cause biopsies with a less severe phenotype than clinical rejection.

## 1. Introduction

Rejection-associated gene expression has been well-characterized in for cause renal allograft biopsies. Three groups of rejection gene transcripts have been differentiated: (a) gene transcripts that are shared by T cell-mediated and antibody-mediated rejection; (b) specific differentially expressed transcripts in T cell-mediated rejection; and (c) specific differentially expressed transcripts in antibody-mediated rejection [1,2,3]. Rejection-associated gene expression is shared among different transplanted organs [4] and is involved in the immune response against infection, cancer, or autoimmune diseases [5]. They are also expressed in surveillance biopsies with subclinical rejection [6], and in biopsies with a histological diagnosis below the threshold of rejection such as borderline changes or interstitial fibrosis and tubular atrophy (IFTA).

Subclinical rejection in surveillance biopsies has been associated in some studies with decreased renal allograft survival [7,8,9]. It has been shown in a study comparing clinical and subclinical rejection that both phenotypes shared differentially expressed genes and pathways. These data suggest that there exists a continuum of alloimmune activation in both situations [10].

The significance of borderline changes is difficult to interpret since it can represent true T cell-mediated rejection (TCMR), acute kidney injury associated with ischemia-reperfusion or other types of tissue injury [11]. Thus, it is not surprising that the histological definition of borderline changes has been modified over the years. In the first Banff meeting, this category was defined as the presence of mild tubulitis (t1) associated with mild to severe interstitial inflammation (i1–i3) [12]. Later, the classification was modified, and biopsies showing mild to severe tubulitis (t1–t3) without interstitial inflammation (i0) were included in this category [13]. Recently, it has been described that patients with biopsy scoring ≥ i1t1 have a poorer outcome than patients with biopsies displaying i0t1 [14], leading to the actual definition of borderline changes as at least i1t1 [15]. Therapeutic approaches towards borderline changes vary in different clinical settings. The diagnosis of borderline changes in biopsies for cause leads to anti-rejection treatment with steroid pulses in most cases, while in many centers the presence of borderline changes in surveillance biopsies is not treated. Thus, borderline changes are considered a heterogeneous diagnosis, ranging from mild inconsequential inflammation that is resolved without a specific treatment to full blown TCMR [16,17,18]. Microarray studies in biopsies for cause displaying borderline changes have shown that most cases designated borderline by histopathology are found to be non-rejection by molecular phenotyping [11]. In the setting of surveillance biopsies, it has been shown that molecular changes of rejection are correlated with histological diagnosis of TCMR or borderline rejection, but this molecular pattern is not associated with graft outcomes [19].

The presence of IFTA in biopsies for cause has been described in association with antibody-mediated rejection, T cell-mediated rejection, glomerulonephritis, and other diseases affecting the graft. On the other hand, IFTA without the presence of a well-defined post-transplant disease is observed in a proportion of patients [20,21]. In surveillance biopsies, IFTA with interstitial inflammation in healthy areas is associated with decreased graft survival when compared with biopsies with IFTA without inflammation [8].

Microarray studies in surveillance biopsies with IFTA have described the presence of rejection-associated transcripts [7]. These observations inspired the definition of chronic T cell-mediated rejection as the presence of moderate to severe inflammation in areas of fibrosis (i-IFTA ≥ 2 and t ≥ 2) [22].

The aim of the present study is to characterize rejection-associated gene transcripts in for cause and surveillance biopsies with subclinical rejection, borderline changes and IFTA.

## 2. Results

### 2.1. Patients

Between July 2015 and August 2018, 435 renal graft biopsies were performed, and 188 biopsies had a third core of tissue stored in the nephrology biobank for microarray studies. Since in 7 patients extracted RNA was of insufficient quality for analysis, 181 patients were considered. Patients with the following histological diagnosis were discarded: chronic pyelonephritis (1), focal and segmental glomerular sclerosis (5), membranous nephropathy (2), IgA nephropathy (6), C3 glomerulopathy (1), polyoma virus BK nephropathy (4) and diabetic nephropathy (1). Histological diagnosis of the considered 161 biopsies is shown in Table 1. Since rejection was an infrequent diagnosis, all cases were included in the microarray study. For the remaining groups, biopsies with the longest follow up were included to complete a total of 96 biopsies (Table 1). Clinical characteristics of the included patients and histological scores are summarized in Table 2 and Table 3.

### 2.2. Principal Component Analysis Using Gene Transcripts

The principal component analysis showed that rejection in indication and surveillance biopsies clustered in the left upper quadrant, normal surveillance biopsies in the lower right quadrant, while borderline changes and IFTA tended to cluster in between (Figure 1).

### 2.3. Rejection-Associated Gene Score (RAG Score)

We described the most differentially expressed genes between biopsies for cause with clinical rejection and normal surveillance biopsies. For this purpose, we adjusted FC and p-value to obtain approximately the 100 most differentially expressed genes between these two groups. A total of 109 differentially expressed genes (*p*-value < 0.01 and fold change (FC) > log 1.75) were obtained as listed in Appendix A. The geometric mean of these 109 differentially expressed genes was calculated to obtain a RAG score.

The RAG score was similar between rejection in biopsies for cause (REJ-C) and subclinical rejection (REJ-S). In borderline changes in biopsies for cause (BL-C), borderline changes in surveillance biopsies (BL-S), interstitial fibrosis and tubular atrophy in biopsies for cause (IFTA-C), and IFTA in surveillance biopsies (IFTA-S) the score was significantly lower than in rejection biopsies. On the other hand, it was significantly higher in BL-C, BL-S, and IFTA-C, than in normal biopsies. There were no differences in the RAG score between IFTA-S and, normal surveillance biopsies (Normal-S) (Figure 2).

To validate this result, we considered the T cell-mediated rejection most differentially expressed genes in kidney transplants as reported by Venner et al. [23], the common rejection module reported by Khatri et al. [4] and the constant of rejection reported by Wang et al. [5]. These three gene sets were identified in our microarray and a rejection score for each of these three gene sets was calculated as the geometrical mean in our sample. We observed that rejection genes described in these studies were significantly higher in rejection than in the other groups. Using the rejection scores obtained from these three studies, we observed that rejection genes in BL-C were significantly higher than in normal biopsies. The rejection scores obtained from Wang et al. and Khatri et al., were significantly higher in BL-C than in IFTA-S. However, there was no difference in the expression of rejection gene score between BL-C, BL-S, and IFTA-C (Figure 3).

### 2.4. RAG Score and Outcome

There was no overlap in the RAG score between REJ-C biopsies and Normal-S biopsies (Figure 2). By receiver operating curve analysis, the best cut off was 5.89 (Youden’s index) with an area under the curve of 1. Accordingly, the studied groups of biopsies were classified as positive (≥5.89) or negative (<5.89) RAG score (Figure 4).

In the 5 study groups, graft outcome (death-censored graft loss or 2-year eGFR deterioration ≥ 30%) was associated with histological diagnosis and with RAG score (Figure 5).

Regarding the control groups, none of the Normal-S group and 6 out of 12 patients with REJ-C reached the composite endpoint.

Multivariate Cox regression analysis showed that RAG score ≥ 5.89 was an independent predictor of graft outcome from histological diagnosis (hazard ratio: 3.5 and 95% confidence interval: 1.1–10.9; *p* = 0.031). Survival analysis excluding patients with subclinical rejection yielded similar results (*p* = 0.004 for univariate and *p* = 0.037 for multivariate analysis).

## 3. Discussion

In the present study, most differentially expressed genes between normal surveillance biopsies and biopsies for cause with rejection, were consistent with rejection-associated gene sets reported by others [4,5,23]. These lists of gene sets have been obtained by different approaches. Venner et al. [23] described the most differentially expressed genes between T cell-mediated rejection and all other diagnoses, including antibody-mediated rejection in biopsies for cause. Khatri et al. [4] employed eight independent data sets from kidney, liver, heart, and lung transplants to characterize the top rejection differentially expressed genes between organs. Wang et al. [5] considered top differentially expressed genes in allograft or cancer rejection, autoimmune disease, and tissue damage during infection. The concordance in the characterization of the top expressed genes during rejection between studies employing different strategies, reflects the stereotyped effector immune response leading to tissue damage triggered by different injuries. In our study, the RAG-score mainly is composed by a set of genes related with antigen presentation, T-cell activation, cytotoxic proteins, chemokine expression, B-cell, and plasma-cell transcripts.

We observed that the RAG-score was not different in surveillance biopsies with subclinical rejection and biopsies for cause with clinical rejection. RAG-score was binarized to classify biopsies as rejection or non-rejection. Only one out of six patients with rejection in surveillance biopsies had a negative RAG-score. This observation suggests that in subclinical rejection, as it has been previously described in clinical rejection [2], there is a reasonable concordance between histological and molecular diagnosis. Furthermore, this observation argues in favor for treating patients with subclinical rejection [18,24,25]

In biopsies for cause with borderline changes or IFTA, and in surveillance biopsies with borderline changes, RAG-score was higher than in normal surveillance biopsies but lower than in biopsies with clinical rejection. A rejection signal was observed in 83% of surveillance biopsies with rejection, 38% of biopsies for cause with borderline changes, 25% of biopsies for cause with IFTA, 17% of surveillance biopsies with borderline changes and 5% of surveillance biopsies with IFTA. The RAG score variability in these groups suggests that there were patients with and without rejection signal in each diagnostic category. Finally, surveillance biopsies with IFTA were not different from normal surveillance biopsies pointing out that stable grafts with IFTA are immunologically quiescent.

The clinical significance of borderline changes is difficult to interpret ranging from true rejection to non-specific inflammation [16]. Lipman et al. [26] showed that inflammatory gene expression in surveillance biopsies was negative in biopsies with normal histology, intermediate in borderline changes, and high in rejection biopsies. Enhanced inflammatory gene expression in surveillance biopsies with borderline changes and rejection was confirmed in a study evaluating early 6-week surveillance biopsies [19]. However, in this study, enhanced inflammatory gene expression was higher in patients with delayed graft function than in patients with immediate function, suggesting that, in early surveillance biopsies, inflammatory gene expression may also reflect injury-repair response. In the other hand, no association between inflammatory gene expression and 2-year graft outcome was observed. However, studies evaluating the utility of histologic diagnosis in early surveillance biopsies to predict outcome have shown that a long follow up is necessary to show an association between early inflammation and graft events [27,28,29,30]. In the present study, RAG-gene score in biopsies with borderline changes was higher in biopsies for cause than in surveillance biopsies. A similar observation was reported in a study comparing early biopsies for cause and 3-month surveillance biopsies [31]. These data suggest that functional deterioration in patients with borderline changes could be partly explained by enhanced rejection associated gene expression. In biopsies for cause with borderline changes, a positive RAG-score, suggests that these cases represent true rejection. This result agrees with previous studies reporting a rejection molecular signature in a similar proportion of cases [11,32].

In patients with IFTA, the RAG-score was positive in one quarter of biopsies for cause and only in 1 out of 20 surveillance biopsies. Graft survival is shortened in surveillance biopsies with IFTA and inflammation in comparison to biopsies with IFTA without inflammation [7,33]. In one-year surveillance biopsies with IFTA and inflammation an overexpression of innate immune transcripts, antigen presentation and cytotoxic T-cell has been described suggesting that mediators of rejection signaling are activated [34]. In another study also evaluating 12-month surveillance biopsies without rejection, biopsies with IFTA expressed macrophage, IFN-gamma, T-cell antigen presentation and T-cell toxicity associated genes [35]. Characterization of differentially expressed genes between normal biopsies and biopsies with IFTA has confirmed that genes related to immune response, inflammation and matrix deposition are overexpressed in biopsies with fibrosis [36]. Furthermore, in a study including for cause and surveillance biopsies, Modena et al. [6] not only described the presence of rejection associated genes in biopsies with IFTA and inflammation, but also in biopsies with IFTA without inflammation. Finally, in another study, it has been observed that there is an overexpression of T cell, IFN-gamma, macrophage and injury-repair transcripts in biopsies with inflammation in healthy areas. By contrast, in biopsies with IFTA and inflammation in scarred areas there is an overexpression of B cells, immunoglobulins, mast cells, and a different set of injury-repair transcripts [37].

In summary, the above-mentioned studies suggest that in patients with IFTA, especially when it is associated with inflammation, there is an over expression of rejection associated genes. In the present study, the inflammatory burden in for cause or surveillance biopsies with IFTA was relatively low. However, a significant proportion of biopsies for cause with IFTA were classified as rejection according to the RAG-score. Altogether our data confirm that there is a discrepancy between histological diagnosis and gene score [38], especially in patients without a histological diagnosis of clinical rejection, i.e., borderline changes and IFTA.

Finally, a positive RAG-score was associated with graft outcome in the study groups. This association was independent from biopsy indication and histological diagnosis. Interestingly, this association was confirmed when patients with subclinical rejection were excluded from the analysis. The main limitation of the study is the lack of a validation cohort to confirm the utility of RAG-score to predict outcome. However, in another study evaluating the utility of a rejection gene score in patients with IFTA without inflammation, either in for cause or surveillance biopsies, there was an association between a high rejection gene score and graft survival [6]. Another limitation is the reduced sample size, especially to evaluate the utility of the RAG-score in surveillance biopsies.

## 4. Materials and Methods

### 4.1. Patients

All adult patients biopsied between July 2015 and August 2018, who gave their informed consent to obtain an additional biopsy core for the nephrology biobank were considered. For the present study, 2 controls and 5 study groups were defined. Control groups were (a) normal surveillance biopsies (Normal-S) and (b) rejection (T cell-mediated and antibody-mediated rejection) in biopsies for cause (REJ-C). The study groups were (a) rejection (T cell-mediated and antibody-mediated rejection) in surveillance biopsies (REJ-S); (b) borderline changes in biopsies for cause (BL-C); (c) borderline changes in surveillance biopsies (BL-S); (d) IFTA in biopsies for cause (IFTA-C); and (e) IFTA in surveillance biopsies (IFTA-S). Clinical T cell-mediated rejection was treated with steroid boluses, active antibody-mediated rejection with plasmapheresis, intravenous immunoglobulins, and rituximab while chronic antibody-mediated rejection was not treated. REJ-S and BL-C were treated according to the attending physician criteria. BL-S and IFTA-C were not treated.

Clinical and demographic characteristics of patients were recorded and anti HLA antibodies at the day of transplant and at the time of biopsy were determined by Luminex technology using the product LIFECODES LifeScreen Deluxe (Gen-Probe-Immucor, Stanford, CT, USA). The present study has been approved by our ethical committee (Comité Etico de Investigación Clínica del Hospital Universitari Vall d’Hebron PR(AG)369/2014, approval date 1 December 2014) and has been performed in accordance with the Declaration of Helsinki, and is consistent with the Principles of the Declaration of Istanbul on Organ Trafficking and Transplant Tourism.

### 4.2. Biopsies

Renal biopsies were performed under ultrasound guidance by trained radiologists with a 16-gauge automated needle. Three cores of tissue were obtained: one was processed for optical microscopy; one was embedded in OCT for immunofluorescence and the other one was stored in RNA later for molecular studies.

The first core was embedded in formalin, paraffin-fixed and 2–4 m thick sections were stained with haematoxylin-eosin, periodic acid Schiff, Masson’s trichrome, and silver methenamine. Histological lesions were evaluated according to Banff criteria [15] and accordingly the definition of borderline changes was: i ≥ 1 and t ≥ 1. All biopsies were stained with an anti-SV40 antibody. Immunofluorescence studies were done in 3-µm cryostat sections stained with FITC-conjugated anti-human IgG, IgA, IgM, C3, C4, lambda and kappa light chains. C4d was stained with indirect immunofluorescence with a monoclonal antibody (Quidel, San Diego, CA, USA) and deposition in peritubular capillaries was graded according to the Banff criteria. The third score was treated with Ambion^®^ RNAlater^®^ Tissue Collection reagent as indicated by the manufacturer and frozen at −80 °C.

### 4.3. RNA Extraction and Microarray Hybridization

Total RNA extraction from renal biopsies was performed by lysing the cells with the TissueLyser II and following the RNAeasy Mini Kit (QIAGEN). RNA quantity and quality were analyzed with the NanoDrop ND2000 (Thermo Scientific, Wilmington, DE, USA) and the Bioanalyzer (Agilent Technologies, Sta. Clara, CA, USA).

Microarrays service was carried out by the High Technology Unit (UAT) at Vall d’Hebron Research Institute (VHIR), Barcelona (Spain), using a GeneTitan^®^ System according to the procedure described by the manufacturer. One plate, containing 96 Clariom S arrays, was used for this experiment. These arrays provide an accurate measurement of the human transcriptome at a gene-level, by using probes covering more than 20,000 well-annotated genes, distributed through constitutive exons.

Briefly, 70 ng of total RNA from each sample were used as starting material. The quality of the isolated RNA was measured previously by capillary electrophoresis (Bioanalyzer 2100, Agilent Technologies, Sta. Clara, CA, USA). Single stranded-cDNA suitable for labeling was generated from total RNA using the WT PLUS Reagent Kit (ThermoFisher Scientific, Lutterworth, UK), according to the manufacturer’s instructions. Purified sense-strand cDNA was fragmented, labeled, and hybridized to the arrays using the GeneTitan Hybridization, Wash and Stain Kit for WT Plates (ThermoFisher Scientific, Lutterworth, UK). The plate was loaded into the GeneTitan and, after array scanning, raw data quality control was performed to check the performance of the whole processing.

### 4.4. Statistics

Results are expressed as raw numbers for categorical variables and as the mean ± standard deviation for continuous variables. Comparison between groups for categorical variables was done by chi-squared test with continuity correction. Comparison between groups for continuous variables was done by analysis of variance (ANOVA) with Scheffé post hoc test for individual comparisons. Kaplan–Meier survival curves with log-rank test and Cox’s proportional hazard model were employed for survival analysis. A composite outcome variable including death-censored graft loss and eGFR deterioration ≥ 30% at 2-years was defined. All *p*-values were two-tailed and *p*-value < 0.05 was considered significant.

Bioinformatic analysis was performed at the Statistics and Bioinformatics Unit (UEB) of the Vall d’Hebron Institute of Research (VHIR, Barcelona, Spain). Robust Multi-array Average (RMA) algorithm [39] was used for pre-processing microarray data. Background adjustment, normalization, and summarization of raw core probe expression values were defined so that the exon level values were averaged to yield one expression value per gene. Selection of differentially expressed genes was based on a linear model analysis with empirical Bayes modification for the variance estimates [40]. To account for multiple testing, *p*-values were adjusted to obtain stronger control over the false discovery rate (FDR), as described by the Benjamini and Hochberg method [41].

Principal component analysis was performed with normalized data from all the genes used in the differential expression analysis. Differentially expressed genes were selected based on a *p*-value < 0.01 and base 2 logarithmic fold change > 1.75. In order to validate our rejection-associated gene set, we calculated in our array the geometric mean of the gene expression of the published list of genes described by Venner et al. [23], Khatri et al. [4], and Wang et al. [5].

## 5. Conclusions

In conclusion, the present data support the utility of the characterization of rejection-associated genes in biopsies with borderline changes, IFTA, and subclinical rejection to improve the risk stratification. These data also raise the question whether characterization of rejection gene expression in biopsies without a clinical and histological diagnosis of rejection may be useful to adjust immunosuppressive treatments and improve outcome.

## Figures and Tables

**Figure 1 ijms-21-08237-f001:**
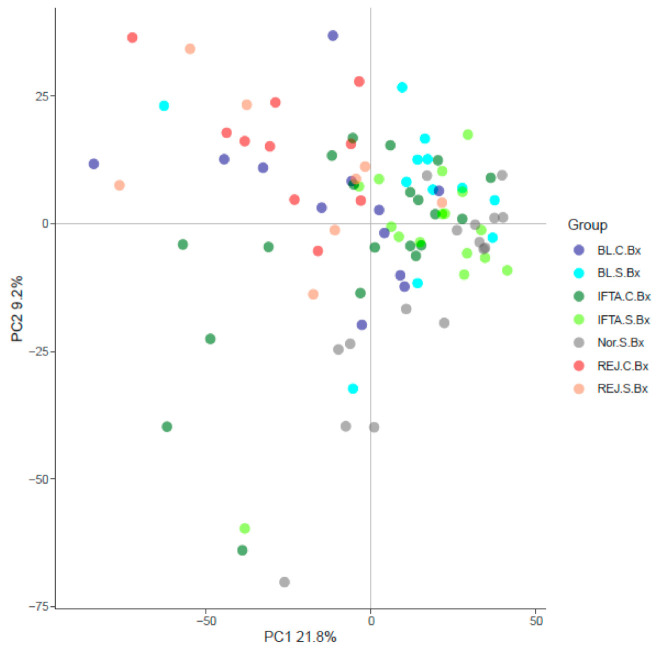
Principal component analysis of the microarray and histological diagnosis in for cause and surveillance biopsies. REJ-C, rejection in biopsies for cause (*n* = 12); Normal-S, normal surveillance biopsies (*n* = 17); REJ-S, rejection in surveillance biopsies (*n* = 6); BL-C, borderline changes in biopsies for cause (*n* = 13); BL-S, borderline changes in surveillance biopsies (*n* = 12); IFTA-C, interstitial fibrosis and tubular atrophy in biopsies for cause (*n* = 20); IFTA-S, interstitial fibrosis and tubular atrophy in surveillance biopsies (*n* = 16).

**Figure 2 ijms-21-08237-f002:**
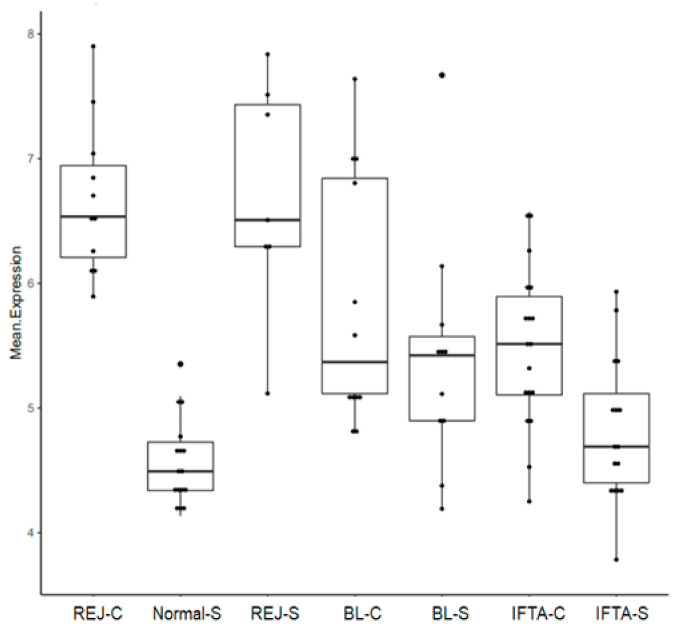
Rejection-associated gene score in the different study groups. REJ-C, rejection in biopsies for cause (*n* = 12); Normal-S, normal surveillance biopsies (*n* = 17); REJ-S, rejection in surveillance biopsies (*n* = 6); BL-C, borderline changes in biopsies for cause (*n* = 13); BL-S, borderline changes in surveillance biopsies (*n* = 12); IFTA-C, interstitial fibrosis and tubular atrophy in biopsies for cause (*n* = 20); IFTA-S, interstitial fibrosis and tubular atrophy in surveillance biopsies (*n* = 16). ANOVA *p*-value 6.27 × 10^−12^.

**Figure 3 ijms-21-08237-f003:**
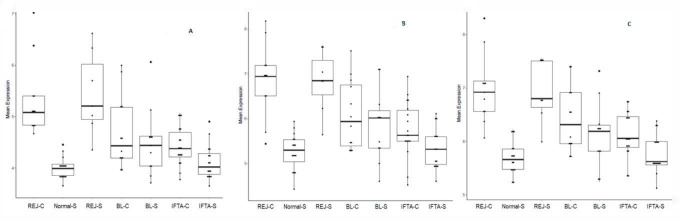
Rejection gene score according to the most differentially expressed genes in Venner et al. [23], Khatri et al. [4], and Wang et al. [5] applied to our set of biopsies ((**A**–**C**) respectively). REJ-C, rejection in biopsies for cause (*n* = 12); Normal-S, normal surveillance biopsies (*n* = 17), REJ-S, rejection in surveillance biopsies (*n* = 6); BL-C, borderline changes in biopsies for cause (*n* =13); BL-S, borderline changes in surveillance biopsies (*n* = 12); IFTA-C, interstitial fibrosis and tubular atrophy in biopsies for cause (*n* = 20); IFTA-S, interstitial fibrosis and tubular atrophy in surveillance biopsies (*n* = 16). ANOVA *p*-value 9.46 × 10^−10^, 2.16 × 10^−9^, 5.08 × 10^−10^ for graphs (**A**–**C**); respectively.

**Figure 4 ijms-21-08237-f004:**
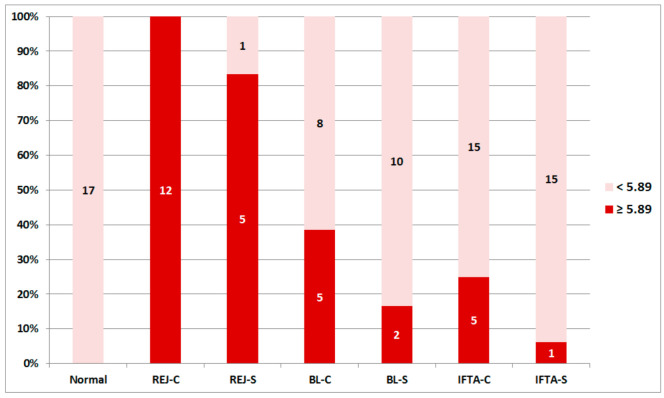
Positive (≥5.89) and negative (<5.89) rejection-associated gene score in the different histological categories in surveillance and for cause biopsies. The number of biopsies in each group according to the rejection-associated gene score is displayed. REJ-C, rejection in biopsies for cause; Normal, normal surveillance biopsies, REJ-S, rejection in surveillance biopsies; BL-C, borderline changes in biopsies for cause; BL-S, borderline changes in surveillance biopsies; IFTA-C, interstitial fibrosis and tubular atrophy in biopsies for cause; IFTA-S, interstitial fibrosis and tubular atrophy in surveillance biopsies.

**Figure 5 ijms-21-08237-f005:**
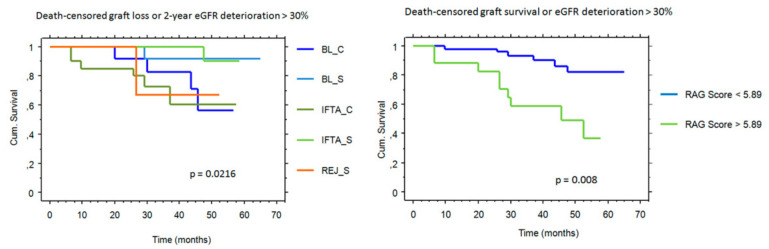
Death-censored graft loss or 2-year eGFR deterioration ≥ 30 % from the date of biopsy in the different histological categories (left panel) and in biopsies with a positive or negative rejection-associated gene (RAG) score (right panel). REJ-S rejection in surveillance biopsies; BL-C, borderline changes in biopsies for cause; BL-S, borderline changes in surveillance biopsies; IFTA-C, interstitial fibrosis and tubular atrophy in biopsies for cause; IFTA-S, interstitial fibrosis and tubular atrophy in surveillance biopsies.

**Table 1 ijms-21-08237-t001:** Available and included for cause and surveillance biopsies.

	Available Biopsies		Included Biopsies	
	For Cause	Surveillance	For Cause	Surveillance
Normal	0	39	0	17
Borderline changes	19	18	13	12
IFTA	35	32	20	16
TCMR	6	5	6 ^a^	5 ^c^
ABMR	6	1	6 ^b^	1 ^d^
Total	66	95	45	51

IFTA, interstitial fibrosis and tubular atrophy; TCMR, T cell-mediated rejection; ABMR, antibody-mediated rejection. ^a^ 3 cases of TCMR grade I and 3 cases grade II; ^b^ 2 cases of active ABMR and 4 cases of chronic active ABMR; ^c^ 2 cases of TCMR grade I and 3 cases grade II and ^d^ 1 case of active ABMR.

**Table 2 ijms-21-08237-t002:** Demographic data for recipients and donors as well as clinical data at the time of biopsy.

Variable	Normal	REJ-C	REJ-S	BL-C	BL-S	IFTA-C	IFTA-S	*p*-Value
N	17	12	6	13	12	20	16	
Age (y)	54 ± 13	43 ± 12	53 ± 21	59 ± 11 ^b^	57 ± 13 ^b,c^	46 ± 15 ^c^	52 ± 17	0.060
Sex (m/f)	12/5	6/6	4/2	8/5	9/3	16/4	10/6	0.610
1st Tx/Re-Tx	17/0	8/4	6/0	13/0	12/0	18/2	12/4	0.002
Donor age (y)	56 ± 15	44 ± 16	59 ± 15	59 ± 14	53 ± 16	58 ± 11	57 ± 16	0.226
HLA (A + B + DR) mm	3.8 ± 1.0	3.7 ± 1.4	4.0 ± 1.9	3.9 ± 1.0	3.8 ± 0.7	3.5 ± 1.3	3.7 ± 1.1	0.942
DGF (no/yes)	15/2	10/1	6/0	5/8	10/2	14/1	13/3	0.010
Rejection (no/yes)	17/0	8/4	4/2	12/1	9/3	19/1	14/2	0.082
Induction (ATG/IL2RAb)	4/13	7/5	2/4	4/15	7/9	4/14	7/9	0.295
Immunosuppression TAC + MMF + S/other	16/1	10/2	4/2	9/4	12/0	14/6	16/0	0.036
Time biopsy (m)	6 ± 5	51 ± 57 ^a^	6 ± 6 ^b^	25 ± 27	7 ± 4 ^b,d^	77 ± 78 ^a,c,e^	7 ± 5 ^b,f^	<0.001
Creatinine (mg/dL)	1.3 ± 0.3	2.0 ± 0.6 ^a^	1.4 ± 0.2	2.5 ± 0.8 ^a,b,c^	1.3 ± 0.3 ^b,d^	2.1 ± 0.7 ^a,c,e^	1.4 ± 0.3 ^b,d,f^	<0.001
UPCR (g/g)	0.4 ± 0.3	1.7 ± 1.5 ^a^	0.3 ± 0.1 ^b^	1.1 ± 1.2 ^a,b,c^	0.3 ± 0.2 ^b,d^	1.3 ± 0.9 ^a,c,e^	0.3 ± 0.3 ^b,d,f^	<0.001
DSA (no/yes)	17/0	8/4	5/1	13/0	12/0	20/0	16/0	<0.001

Tx, transplant; HLA mm, HLA mismatches at the loci A + B + DR; DGF, delayed graft function; TAC + MMF + S, tacrolimus associated with mycophenolate and steroids; UPCR, urinary protein to creatinine ratio; DSA, HLA donor specific-antibodies determined by Luminex technology. ^a^
*p* < 0.05 vs. normal, ^b^
*p* < 0.05 vs. REJ-C, ^c^
*p* < 0.05 vs. REJ-S, ^d^
*p* < 0.05 vs. BL-C, ^e^
*p* < 0.05 vs. BL-S, ^f^
*p* < 0.05 vs. IFTA-C by Scheffé test.

**Table 3 ijms-21-08237-t003:** Histological Banff scores.

Variable	Normal	REJ-C	REJ-S	BL-C	BL-S	IFTA-C	IFTA-S
N	17	12	6	13	12	20	16
Glomeruli (N)	16 ± 8	16 ± 6	17 ± 5	17 ± 6	21 ± 11	17 ± 8	23 ± 12
Gs (%)	7 ± 7	22 ± 23	12 ± 16	22 ± 15	12 ± 14	31 ± 24	10 ± 6
g	0.1 ± 0.3	1.4 ± 1.1	0.8 ± 1.3	0.2 ± 0.4	0.4 ± 0.8	0.3 ± 0.6	0.2 ± 0.4
i	0.1 ± 0.2	1.3 ± 1.1	1.5 ± 0.5	1.2 ± 0.7	0.7 ± 0.6	0.1 ± 0.3	0.1 ± 0.2
t	0	1.2 ± 0.9	1.8 ± 0.7	1.0 ± 0	1.0 ± 0.6	0.2 ± 0.4	0.1 ± 0.3
v	0	0.3 ± 0.6	0.5 ± 0.5	0	0	0	0
ah	0.2 ± 0.4	1.4 ± 1.2	0	1.0 ± 0.9	0.7 ± 0.-6	1.3 ± 1.2	0.6 ± 0.6
cg	0	0.7 ± 0.9	0	0	0.2 ± 0.6	0.2 ± 0.5	0
ci	0.1 ± 0.3	1.2 ± 0.9	0.8 ± 1.0	1.3 ± 0.6	0.9 ± 0.8	1.8 ± 0.8	1.3 ± 0.5
ct	0.6 ± 0.5	1.1 ± 0.8	1.0 ± 0.6	1.2 ± 0.4	1.0 ± 0.4	1.6 ± 0.9	1.1 ± 0.3
cv	0.4 ± 0.7	0.8 ± 0.8	0.3 ± 0.5	1.0 ± 0.8	0.4 ± 0.5	1.3 ± 1.2	0.9 ± 0.8
mm	0.1 ± 0.2	0.5 ± 0.7	0	0.1 ± 0.3	0	0.2 ± 0.7	0
ptc	0.1 ± 0.2	1.4 ± 0.8	0.7 ± 1.2	0.6 ± 0.8	0.4 ± 0.8	0.5 ± 0.8	0.1 ± 0.2
C4d	0	0.3 ± 0.5	0	0	0	0	0
i-IFTA	0.4 ± 0.9	2.1 ± 1.3	2.2 ± 1.3	2.1 ± 0.8	1.3 ± 1.1	2.2 ± 1.0	1.8 ± 0.9
t-IFTA	0.2 ± 0.4	0.9 ± 0.6	0.8 ± 0.4	0.7 ± 0.7	0.6 ± 0.7	0.7 ± 0.6	0.7 ± 0.5
i-total	0.1 ± 0.2	1.3 ± 0.6	1.7 ± 0.6	1.5 ± 0.7	0.8 ± 0.6	0.7 ± 0.5	0.4 ± 0.3

Gs, percentage of globally sclerosed glomeruli; g, glomerulitis; i, interstitial infiltrate; t, tubulitis; v, endothelialitis; ah, arteriolar hyalinosis; cg, transplant glomerulopathy; ci, interstitial fibrosis; ct, tubular atrophy; cv, intimal thickening; mm, mesangial matrix increase; C4d, deposition of C4d in peritubular capillaries; i-IFTA, inflammation in areas of interstitial fibrosis; t-IFTA, tubulitis in atrophic tubules.

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
