# Peer review of "A Rejection Gene Expression Score in Indication and Surveillance Biopsies Is Associated with Graft Outcome"

_ijms, 2020, doi:10.3390/ijms21218237_

Round 1
Reviewer 1 Report
This manuscript correlates a score based on rejection genes expressed in renal allograft biopsies (RAG) to the pathological findings and clinical course. The paper is overall well written although the continual use of abbreviations takes some getting used to.
As a clinician, my biggest methodological questions are the timing of the biopsies relative to the age of the allograph as one might expect some mild chronic changes in a 10-year-old allograph compared to one in place for a year. Similarly, in figure 5 is the survival data from the time of the biopsy or from the time of the transplant? I do think inclusion of a larger number of the IFTA and borderline patients in the data presented might be informative, as long as they have at least a 2 year follow-up since the small number of patients in each group is the major weakness of the paper.
Much of the discussion talks about inflammatory genes, but mention is made of immune function genes. In the introduction or in the discussion it would be interesting to learn what classes of genes make up the RAG. I know the genes are identified in the supplementary material, but the function of each gene is not present.
Specifics:
Table 3 perhaps should be in the supplemental material
Figure 3 is basically unreadable as it is very small
In figure 1 putting the color ID’s in the footer as well as alongside the figure would be helpful. Also, I assume ‘REB’ should be ‘REJ’.
Figure 2: The number of biopsies should be stated
Figure 4 25.89??
Author Response
This manuscript correlates a score based on rejection genes expressed in renal allograft biopsies (RAG) to the pathological findings and clinical course. The paper is overall well written although the continual use of abbreviations takes some getting used to.
We have reduced the number of abbreviations from 10 to 8
As a clinician, my biggest methodological questions are the timing of the biopsies relative to the age of the allograft as one might expect some mild chronic changes in a 10-year-old allograft compared to one in place for a year. Similarly, in figure 5 is the survival data from the time of the biopsy or from the time of the transplant? I do think inclusion of a larger number of the IFTA and borderline patients in the data presented might be informative, as long as they have at least a 2 year follow-up since the small number of patients in each group is the major weakness of the paper.
In figure 5, the survival from the time of the biopsy is described. This point has been specific in the legend of figure 5.
Much of the discussion talks about inflammatory genes, but mention is made of immune function genes. In the introduction or in the discussion it would be interesting to learn what classes of genes make up the RAG. I know the genes are identified in the supplementary material, but the function of each gene is not present.
In the discussion a sentence has been added to describe the main function of the genes included in the RAG score.
Specifics:
Table 3 perhaps should be in the supplemental material.
We prefer to leave table 3 in the main text since it is important for the reader to be able to check the severity of the individual lesions in the different diagnostic groups
Figure 3 is basically unreadable as it is very small.
Figure 3 has been enlarged
In figure 1 putting the color ID’s in the footer as well as alongside the figure would be helpful. Also, I assume ‘REB’ should be ‘REJ’.
This was a mistake that has been corrected in the new version.
Figure 2: The number of biopsies should be stated
The number of cases in each diagnostic category is specified in the legend of figure 2
Figure 4 25.89??
Figure 4 has been corrected in order to better display this number (≥ 5.89).
Reviewer 2 Report
The topic is an important clinical question. The paper would benefit from a power analysis to better describe how to apply the results.
The paper is out of order from standard practice. The Results appear before the Materials and Methods, and the Statistics. It needs to be in a more standard "IMRAD" format.
The discussion is extensive, but not easy to read. It should be streamlined slightly. It would also benefit from a simple discussion of the limitations of the paper.
Author Response
The topic is an important clinical question. The paper would benefit from a power analysis to better describe how to apply the results.
The present investigation was exploratory. We defined the RAG score using normal and rejection biopsies and we evaluated using a full plate microarray what is the expression in a different set of biopsies displaying lesions not fulfilling rejection criteria according to the Banff criteria, e.g., borderline changes and IF/TA either observed in stable grafts (surveillance biopsies) or in deteriorating grafts (biopsies for cause). Additionally, the RAG score is associated with graft outcome independently of histological diagnosis. Of course, we understand that the present results should be evaluated in a large set of biopsies and we have planned to do it in new studies.
The paper is out of order from standard practice. The Results appear before the Materials and Methods, and the Statistics. It needs to be in a more standard "IMRAD" format.
We have checked this point and we follow the instruction of IJMS (Introduction, Results, Discussion and Patients and Methods).
The discussion is extensive, but not easy to read. It should be streamlined slightly. It would also benefit from a simple discussion of the limitations of the paper.
We have simplified the discussion according to your suggestion.